# Obesity and Neurocognitive Performance of Memory, Attention, and Executive Function

Antonio G. Lentoor

Department of Clinical Psychology, School of Medicine, Sefako Makgatho Health Sciences University, Ga-Rankuwa, Pretoria 0208, South Africa; dr.lentoor.antonio@gmail.com; Tel.: +27-(0)-125214767

**Abstract:** Background: Obesity has been linked to an increased risk of dementia in the future. Obesity is known to affect core neural structures, such as the hippocampus, and frontotemporal parts of the brain, and is linked to memory, attention, and executive function decline. The overwhelming majority of the data, however, comes from high-income countries. In undeveloped countries, there is little evidence of a link between obesity and neurocognition. The aim of this study was to investigate the effects of BMI on the key cognitive functioning tasks of attention, memory, and executive function in a South African cohort. Methods: A total of 175 females (NW: BMI = 18.5–24.9 kg/m$^2$ and OB: BMI > 30.0 kg/m$^2$) aged 18–59 years (M = 28, SD = 8.87 years) completed tasks on memory, attention, and executive functioning. Results: There was a statistically significant difference between the groups. The participants who had a BMI corresponding with obesity performed poorly on the tasks measuring memory ($p = 0.01$), attention ($p = 0.01$), and executive function ($p = 0.02$) compared to the normal-weight group. Conclusions: When compared to normal-weight participants, the findings confirm the existence of lowered cognitive performance in obese persons on tasks involving planning, decision making, self-control, and regulation. Further research into the potential underlying mechanism by which obesity impacts cognition is indicated.

**Keywords:** body mass index; brain function; cognition; developing context; neuropsychological tests; obesity

## 1. Introduction

Obesity is a significant public health problem that contributes to the overall burden of disease globally [1]. Obesity is an excess of fat mass caused by an imbalance in energy intake and energy expenditure [2], and a complex interplay between genes and the environment [3]. Obesity is a risk factor for multiple health issues, such as diabetes and hypertension, and a major cause of premature mortality. Body mass index (BMI) given by dividing the weight by height (kg/m$^2$), is a cost-effective marker of identifying a person as obese (BMI above 30 kg/m$^2$) and has been extensively used as a proxy measure for adiposity. The enormous health and financial burden associated with obesity makes it an important research topic [4]. While the physical health sequelae of obesity are well understood, recent empirical evidence suggests a significant impact on the brain [5]. Obesity is a risk factor for neurodegenerative changes and has a deleterious impact on brain function and structure [6]. Atrophy of the temporal brain region, hippocampal, and frontal structure have been found in obese individuals [7]. While the mechanism by which obesity is associated with cognitive function requires furthers explication, pathophysiological changes including oxidative stress, metabolic changes, neuroendocrine dysregulation, and systematic neuroinflammation have been suggested as key mechanisms in hippocampal and frontostriatal dysfunction in obesity [8–10]. Particularly, adipokines, including leptin, interleukin (IL-6), and tumour necrosis factor (TNF-α), have been linked to a weakened blood–brain barrier and obesity-related brain dysfunction [7,8]. Changes in the function and structures of the brain include cognitive problems in memory, attention, and executive function. Memory, which is the ability to store, maintain, and retrieve information,

relies on the integrity of the hippocampal structure [11]. There is increasing evidence from obesity-cognition research to suggest that episodic memory has an important role in regulating consumption in humans [12]. For example, memory of recent meals showed to have an impact on long-term satiation of meals, while both executive function and episodic memory is crucial for the regulation of consumption (i.e., control of food intake) in obese individuals [13]. On the other hand, attention and executive function, which involve the skills of regulation, control, decision making, and appropriate behavioural responses, is largely mediated by the frontostriatal regions of the brain [14].

Evidence from neuropsychological studies suggests that early adult and middle-aged obese individuals compared to normal-weight people showed lower performance on higher-order cognitive function tasks [15]. Several studies found that obese individuals have a lowered performance on executive tasks of planning, problem solving, and cognitive flexibility when compared to normal-weight individuals [8]. Coinciding with these findings is the observation that obesity negatively impacts memory. Cheke and colleagues [12] found that a higher BMI was associated with significantly lower performance on episodic memory tests when compared with individuals with a normal BMI. Likewise, Nguyen and colleagues showed that obesity is associated with poor performance on a task measuring short-term memory that is essential for comprehension, learning, and planning [16].

Obesity [17] and cognitive disorders [18] are substantial contributions to the worldwide burden of non-communicable disease disability, especially in the sub-Saharan African (SSA) region. In South Africa, like in other SSA regions [19], the burden of obesity is disproportionately higher in women than males [17]. Because of the shifting socioeconomic situation in post-apartheid South Africa, women now have access to higher education, are financially secure enough to acquire and purchase high-calorie-density meals, and work in positions that limit physical activity [17]. Evidence also suggests that weight increase throughout reproductive years is a key cause of obesity in women [20]. It is worth noting that African women of reproductive age are more likely to use injectable hormonal contraceptives such as Depo-Provera, which has been linked to a considerable gain in body fat and eventually obesity [21]. Depo-Provera was linked to changes in menstruation pattern, bone mineral density loss, considerable weight gain, and higher BMI in Ethiopian women who took it compared to non-users, according to research done in Northwest Ethiopia [22]. Depo-Provera was related in another study to causing alterations in the hypothalamus appetite control centre [23], which led to weight gain in Depo-Provera users and was linked to increased food consumption associated with higher hunger and, as a result, more carbohydrate-rich diets.

Obesity has been associated with a number of physiological and metabolic changes [24] that could possibly act as pathways for underlying neurodegeneration and dementia risk in later life. Endocrine functional alteration activated by hormonal changes and adipokines due to increased BMI may have a role in the development of increased inflammation and cardiovascular changes that may result in alterations in brain structure, blood–brain barrier integrity, and atrophy which increases vulnerability to neurodegeneration [25].

While obesity and cognitive disorders have grown considerably more common in developing countries, the bulk of existing obesity-cognition neuroscience research is from developed countries, making direct comparisons to the developing setting challenging. In South Africa, there is a dearth of studies on obesity, including its correlates. If we can understand the relationship between obesity and cognitive vulnerability, it can help us take the necessary steps towards developing interventions that can prevent cognitive decline and reduce the risk of dementia in the future. Hence, the current study investigated the association between obesity and cognition, with a specific focus on memory, attention, and executive function, in a cohort of women in a developing context. Based on existing research findings [11,15], we expected to find differences in the neurocognitive performance of obese and normal-weight women. More specifically, we expected to find significantly lower neurocognitive performance scores on the domains of memory, attention, and executive function for obese than normal-weight women.

## 2. Materials and Methods

### 2.1. Study Design and Participants

In this cross-sectional, quantitative study, a total of 308 participants were screened for potential participation, while 133 individuals were excluded (Figure 1). A total of 175 females, aged 18–59 years (28 + 8.87 years) with normal (NW: BMI = 18.5–24.9 kg/m$^2$) or obese (OB: BMI $\geq$ 30.0 kg/m$^2$) weight were enrolled into the study through purposive sampling techniques from a local hospital, university, and community between January and December 2018 via poster advertisements and word of mouth, in the northern parts of Gauteng Province. The community from which the participants were recruited can be classified as peri-urban, with access to public transport, an academic hospital, schools, universities, malls, and other services.

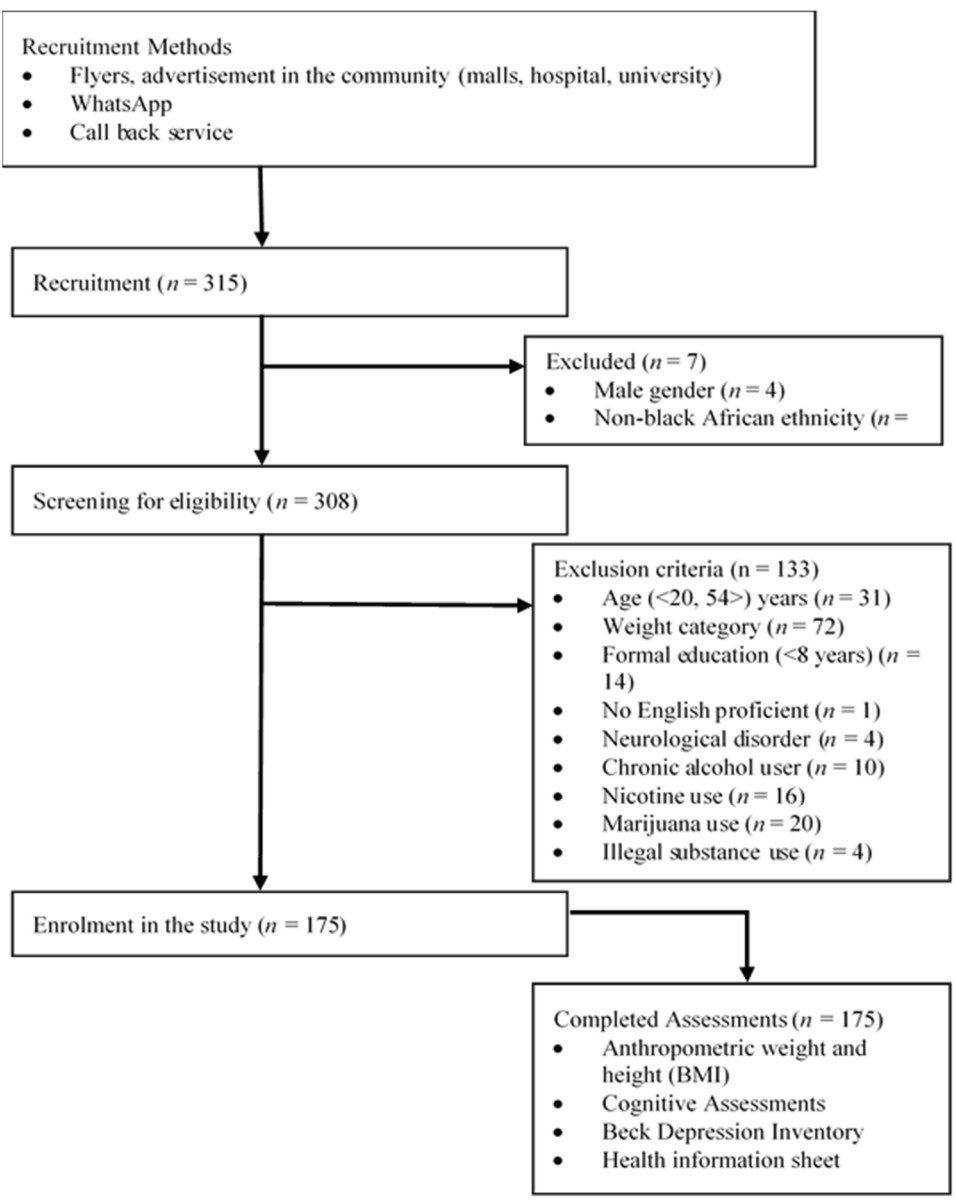

**Figure 1.** Flow chart of the study recruitment procedure.

### 2.2. Eligibility Criteria

Women, according to Nglazi and Ataguba [17], bear a disproportionate burden of obesity as compared to males. Based on a report from the South African health ministry, over 41% of women and 11% of males aged 15 and above were obese in 2019. Therefore, this study recruited females aged 18 to 59 years. Volunteers who responded were initially

screened for eligibility, with those eligible for inclusion reporting no psychiatric and neurological conditions and no use of medication/substances known to alter mood and cognitive capacity that could impact on neurocognitive performance; a proficiency in English to be able to complete the neurocognitive assessment; not being younger than 18 or older than 59 years; and no visual, hearing, or motor coordination problems that would restrict them from completing the assessments.

### 2.3. Data Collection

After completing a screening, all participants meeting the inclusion criteria completed an anthropometric assessment, followed by a neurocognitive functioning assessment that was conducted by a research assistant who was a trained graduate-level student psychologist, and confirmed by the author who is a licenced clinical psychologist trained in neuropsychology. Written informed consent was obtained from all the participants prior to the completion of the assessments. All tests were conducted in a quiet and private space to avoid any distractions and noise interferences.

### 2.4. Measures

A trained graduate-level student psychologist, under the supervision of a licenced clinical psychologist, administered all the assessments. The participants completed domain-specific neurocognitive assessments [26–28].

#### 2.4.1. Sociodemographic and Health Information Sheet

The participants completed an information sheet on age, gender, ethnicity, and health comorbidities (i.e., diabetes, hypertension). The 21-item Beck Depression Inventory-III (BDI-III) [29,30] was administered to screen for depression. A score of 0–9 indicates no depression, while a score of 10 and above is indicative of depression. The BDI-III is a reliable and valid tool that has been use cross-culturally.

#### 2.4.2. Anthropometry

BMI was calculated as the ratio of weight (in kilograms) to height (in metres) squared. Height was measured to the nearest 0.1 cm with a measuring tape. Weight was measured to the nearest 0.1 kg (kilogram) on a digital scale. A BMI of 18.5–24.9 $kg/m^2$ was considered normal weight, and a BMI $\geq$ 30.0 $kg/m^2$ obese.

#### 2.4.3. Memory

The research assistant read out five words with unrelated meaning. Two learning trials of the five words were administered [31]. To assess short-term memory, the participants recalled the words immediately. After approximately 10 min, the participants recalled the five words. The total score of five for delayed memory was assigned. Points were given for uncued recall only.

#### 2.4.4. Attention

The Digit Span subtest has two tasks in which the participant recites five digits forward and recites three digits in a reverse order [31]. The participant's score was the sum of the two tasks, with a total score of two being obtainable if performed correctly.

#### 2.4.5. Executive Function

Multiple aspects of executive functions were assessed using the alternating task adapted from the trail-making B task, the three-dimensional cube copy, and the clock drawing test subtest [32]. The scoring methods for all tests were based on the MoCA scoring system as a combined score of executive function.

The Trail-Making Test (TMT), in which the participant has to connect a set of dots as quickly and accurately as possible, was administered [33]. TMT Part B, which requires participants to connect numbers and letters in alternating sequence, was completed. The

participant was given 1 point if completed correctly. TMT measured executive function, in addition to attention, self-regulation, visual speed and processing and mental flexibility, and task shifting.

The participants were told to copy the three-dimensional cube figure from the example, 1 point was given if copied correctly and 0 if incorrect [34].

The Clock Drawing Test (CDT) draws on different elements of a clock, including the clock face, numbers, and arrow-hands [35]. The participants were asked to draw a clock and to place the clock hands to read '10 past 11'. Points are given for numbers placed in the correct position, for the accuracy of the hands denoting the time 11:10, and contour (total score of 3 points if correct). Executive skill demands made of the CDT include planning (goal-directed behaviour), inhibitory control, and cognitive shifting (adopting a new strategy).

### 2.5. Statistical Analysis

All data were coded and analysed using SAS 9.3 (SAS Institute Inc., Cary, NC, USA). The data are presented as means $\pm$ standard deviations (SD) for continuous variables and frequencies for categorical variables. The inter-group comparison was performed using the chi-square test, and a Student's *t*-test for independent variables to compare OB and NW groups for anthropometric parameters, memory, attention, and executive function. A *p*-value < 0.05 (two-tailed) was considered statistically significant.

## 3. Results

### 3.1. Descriptives

Table 1 presents the descriptive statistics of the sample of participants in this study. The sample included 175 obese ($n = 75$) and normal-weight ($n = 100$) women. A statistically significant difference was found for age ($p < 0.001$), with the obese group being older (mean age = 33.76 years). There was a statistically significant difference in health co-morbidity ($p < 0.001$) between the normal-weight and obese group, as shown in Table 1. The obese group and normal-weight group did not differ on emotional adjustment ($p = 0.07$).

**Table 1.** Descriptive data of the study sample.

| Characteristics | Normal-Weight Group ($n = 100$) | Obese Group ($n = 75$) | *p*-Value |
|---|---|---|---|
| Age | | | |
| Mean | 23.73 | 33.76 | <0.001 |
| SD | 4.08 | 10.21 | |
| Gender | | | |
| Female (%) | 57.14 | 42.86 | 0.000 |
| Health comorbidity (%) | | | |
| Yes | 7.14 | 92.86 | <0.001 |
| No | 61.49 | 38.57 | |
| Depression | | | |
| Mean scores | 8.93 | 10.88 | 0.07 |
| SD | 7.51 | 9.38 | |
| BMI (kg/m$^2$) | | | |
| Mean BMI | 22.27 | 39.46 | 0.000 |
| SD | 1.91 | 9.17 | |

### 3.2. Association between Cognitive Domain Tasks and Body Mass Index (BMI)

A significant inverse association was found between BMI and performance on the cognitive test of delayed recall ($r = -0.19$, $p = 0.001$), clock drawing test ($r = -0.15$, $p = 0.01$), cube copying test ($r = -0.16$, $p = 0.005$), serial 7's test ($r = -0.20$, $p = 0.000$), and digit span test ($r = -0.13$, $p = 0.02$) (Table 2).

**Table 2.** Association between body mass index and neurocognitive test performance.

| Neurocognitive Test | BMI ($n$ = 175) | |
|---|---|---|
| | $r$ | $p$-Value |
| Memory-delayed Recall | −0.185 ** | 0.001 |
| Clock Drawing Test | −0.145 * | 0.010 |
| Cube Copy | −0.158 ** | 0.005 |
| Serial 7s | −0.202 ** | 0.000 |
| Digit Span Test | −0.133 * | 0.018 |

BMI: Body Mass Index, Significant $p$-values * $p$ < 0.05, ** $p$ < 0.01.

### 3.3. Differences in Neurocognitive Performance for Normal-Weight and Obese Groups

Table 3 reports the results of the performance on cognitive tasks for both the normal-weight and obese group. The Student's $t$-test results showed statistically significant differences between the obese and normal-weight groups. Participants' performance on cognitive tasks of attention ($t$(173) = 2.39, $p$ = 0.01), memory, ($t$(173) = 2.52, $p$ = 0.01), and executive function ($t$(173) = 2.31, $p$ = 0.02) was significantly lower in the obese group (OB) as compared to the normal-weight group (NW). The lower means scores obtained by the obese individuals indicate their difficulties in each of the tasks on the cognitive domains.

**Table 3.** Neurocognitive functioning performance by weight group.

| Characteristics | Normal-Weight Group | | Obese Group | | 95% CI | |
|---|---|---|---|---|---|---|
| | Mean | SD | Mean | SD | $t$ | $p$-Value |
| Executive function | 3.67 | 1.15 | 3.25 | 1.20 | 2.31 | 0.02 |
| Attention | 5.41 | 0.88 | 5 | 1.27 | 2.38 | 0.01 |
| Memory | 3.74 | 10.21 | 3.2 | 1.49 | 2.52 | 0.01 |

## 4. Discussion

The present study assessed performance in neurocognitive function with a focus on memory, attention, and executive function in obese and normal-weight individuals.

Similar to Farooq and colleagues [11], who assessed the cognitive performance tasks of attention, memory, and planning executive function of 220 women on the Cambridge Neuropsychological Test, this study found a significant association between the cognitive subdomains and BMI. The study found that body mass index (a proxy measure for adiposity) correlated with several cognitive tests: memory-delayed recall ($p$ = 0.001), clock drawing test ($p$ = 0.010), cube-copying ($p$ = 0.005), serial 7's ($p$ = 0.000), and digit span test ($p$ = 0.018). The tests are known to tap into cognitive domains of attention, memory, and executive function, reflecting frontostriatal functionality. These tests are the most well-researched neuropsychological tests, and they can detect a wide variety of neurodegenerative illnesses, such as mild cognitive impairment and dementia [28]. Similar to the current findings, Fergenbaum et al. [32] found that BMI was associated with poor cognitive performance on the clock drawing test and trail making test. Likewise, poor performance on the digit span test, as a measure of executive function, and the memory-delayed recall test were found to be associated with a high BMI [28,31]. The inverse association between BMI and performance on these cognitive tests could reflect the underlying impact of increased adipose on cognition.

This study found a significant difference in cognitive performance on memory, attention, and executive function tasks, in terms of BMI—a finding supported in several previous studies [11,12,36]. The performance on the memory task was significantly lower for the participants who had a high BMI, corresponding with obese individuals, compared to individuals with a normal weight in this study. Previous research [37] found that individuals with a higher BMI had greater hippocampal atrophy, which is expressed as a reduction in hippocampus volume and is associated with deficits in memory. Similar to

Yang and colleagues [31], this study showed that obese participants performed significantly worse on the modified digit span backwards working memory task. This is an important finding since lower working memory has significant clinical implications. Working memory plays an important role in facilitating adherence to weight management programs. Lower working memory in obesity has been linked to a lower ability to keep goal-relevant information in mind. Previous research [38] found memory deficits in obese individuals was linked to poor appetite control that was associated with lowered orbitofrontal cortical volumes. Furthermore, Yang and colleagues [31] found that a lower performance on working memory tasks in individuals with a higher BMI was associated with increased levels of inflammatory protein (CRP). This supports the role of neuroinflammation in the relationship between obesity and working memory dysfunction [10]. This viewpoint was corroborated by a recent analysis including a decades' worth of epidemiological data, which revealed that adipokine activation in adipose tissue may alter brain function [39]. Adipokines, such as peripheral leptin, interleukin-6 (IL-6), and tumour necrosis factor alpha (TNF-$\alpha$), are released by adipose tissue and interact directly with certain nuclei, including the hippocampus. Adipokines have been linked to obesity-related inflammation and direct energy metabolism dysfunction. Not only does this assist in the regulation of eating behaviour, but it also supports issues with memory consolidation in obese people, which has been linked to cognitive dysfunction, including learning and memory processing difficulties, and risk of dementia and Alzheimer's disease [6].

In contrast to Gunstad [40] and Farooq and colleagues [11], this study found that obese individuals performed significantly lower on the attention task relative to normal-weight individuals. The finding is similar to Tsai and colleagues who showed that obesity was associated with the reduced modulatory ability of attentional networks [41]. Similar to this study, Cook and colleagues [42] investigated the relationship between obesity and cognitive function in 299 women aged 18–35 years and found a significantly lower performance on attention tasks for obese compared to normal-weight individuals.

As expected, obese individuals performed poorly on executive tasks compared to individuals with a normal weight. This finding is consistent with previous studies [43] that showed higher BMI was associated with executive dysfunction (see review by Yang and colleagues). The current findings are also supported by Dassen and colleagues [44] who showed that obese individuals had a lower performance on executive function measures relative to normal-weight individuals. The researchers found that obese individuals reported a poorer performance on neurocognitive tasks specific to inhibition and self-regulation. Similar to the current findings, a cross-sectional study in Canada assessed neuropsychological function using the Clock Drawing Test and the Trail Making Test as a measure of executive function and found that obese participants were almost fourfold more likely to show poor executive function compared to non-obese individuals [32].

This study finding has important clinical implications. Executive function, which is largely mediated by frontal brain regions, plays an important role in self-regulation, behavioural inhibition, shifting, and goal-directed behaviour [13]. These are important cognitive skills for weight-related behaviours. People with a high BMI (a proxy for obese) were shown to have functional and structural connectivity abnormalities in the frontostriatal system [45]. Importantly, abnormalities in this neuroanatomical system are related to deficits in cognitive flexibility (i.e., a key cognitive control function). For example, a deficit in cognitive flexibility, self-regulation, and inhibitory control has been shown to be associated with higher food intake and less exercise and thus increases the risk for obesity [45]. Likewise, research has shown that persons with obesity exhibit less efficient general and food-specific behavioural inhibition, which is linked to weight control problems [46]. Impairment in shifting ability is an important facet of executive function and has been shown to be associated with pathological eating behaviour in individuals with obesity [47]. Importantly, while obesity and cognitive deficiencies are linked, their influence appears to be restricted to early adulthood, with contradictory findings in older adult populations. Moreover, lowered cognitive performance in midlife obesity has been linked to an increased

risk of neurodegenerative pathologies such as Alzheimer's disease [16] and other dementias, emphasizing the need for more research that could help to establish if early interventions and lifestyle modifications targeting at-risk groups can reduce future dementia risk.

This study found no significant difference in the emotional functioning between high-BMI and normal-BMI individuals. Previous studies provide evidence of a link between BMI and depression risk [48]; however, in some studies the link was confined only to individuals with severe obesity. It is important to note, when trying to understand this inconsistency in research, that in many African cultures overweight or large body size has been associated with richness, health, strength, and fertility [49]. Perhaps the cultural perception of body image and body size perception is an important mediator of depression risk.

Importantly, in this study participants with a high BMI were significantly older. There was an average age difference of 10 years between participants with a high BMI, corresponding with obesity (age range 20–54 years), and those with a normal BMI (age range 20–41 years). Additionally, the majority (>90%) of the obese individuals reported cardiometabolic disease comorbidities such as diabetes and hypertension. This finding backs up earlier studies that showed a link between a higher BMI and an increased risk of hypertension and diabetes, with the risk peaking between the ages of 18 and 53 [50]. Previous research found that age and metabolic disorder, both independently and in combination, may increase the risk for cognitive decline in obese individuals, especially in midlife [51].

This study builds on the existing evidence that obesity impacts performance on cognitive tasks related to memory, attention, and executive function; however, the exact underlying mechanism by which obesity affects cognition is far from clear.

Important to note, data on the subjective cognitive function or self-reported daily functioning capacity of the participants was not collected in this study. It is common practice for women in this community to be the primary caretakers of the family while still juggling fulltime employment. These roles require them to draw on higher order cognitive functioning skills on a daily basis. It would therefore be important in future studies to reconcile these roles and responsibilities with performances on standard cognitive tests to get a better understanding of cognitive testing results. This is an important limitation of the study that must be kept in mind when interpreting the findings of this study. There is considerably more evidence from cognitive neuroscience work that cultural experiences and behavioural practices affect neural structure and function [52]. It is critical that the meaning of culture in the context of neuropsychological testing be properly stated in neurocognitive science research. Focusing on the fundamental neuroscience underlying how different aspects of culture influence cognitive test performance and how it relates to brain function will only improve study findings. Another limitation of this study is the cross-sectional design that precludes drawing causative inferences. Third, the use of the BMI metric as the only parameter of adiposity was also a limitation of this study. Fourth, the study used a small purposively selected community-based sample which therefore may not be representative; as such, the findings must be interpreted with caution. Finally, while the brief neurocognitive tests utilized in the study are often used in clinical practice in South Africa, they are not normalized for this population. The use of a detailed neuropsychological battery with a cross-cultural focus could ameliorate this limitation in future studies. The absence of self-reported functioning signals a cautious interpretation of the findings. Future comparative study designs could include the participants' perception of their personal functional state to enhance the findings.

## 5. Conclusions

Overall, in this study individuals with a higher BMI, corresponding with obesity, performed poorly on tasks of memory, attention, and executive function relative to normal-weight individuals. Poor performance on these tasks reflects overall deficits in these neurocognitive domains. The cognitive performance of obese individuals could reflect the functional and structural brain changes that form part of the pathophysiological effects of obesity on the brain. Because cognitive capabilities are considered predictors of eating

and body-weight behaviour change, the findings of this study have substantial treatment implications. Additionally, because obesity is a modifiable risk factor for cognitive decline, dementia, and Alzheimer's disease, early identification and management are more likely to minimize the chance of getting these diseases later in life. Adults with a high BMI, corresponding with obesity, may benefit from interventions designed to lower the risk of cognitive loss. The study highlighted important limitations that could be addressed in future studies. Further knowledge on the underlying mechanism via which obesity impacts the brain is needed. The current study findings could be enhanced in future longitudinal studies that combine objective and subjective cognition, neuroimaging, and biological markers with a relatively large cohort reflecting the diverse population of the context.

**Funding:** This research received no external funding.

**Institutional Review Board Statement:** The study was conducted in accordance with the Declaration of Helsinki and was approved by the Institutional Ethics Committee of Sefako Makgatho Health Sciences University.

**Informed Consent Statement:** Informed consent was obtained from all subjects involved in the study.

**Data Availability Statement:** The data presented in this study are available on reasonable request from the corresponding author. The data are not publicly available due to ethical reasons related to subjects' confidentiality.

**Acknowledgments:** The subjects are acknowledged for voluntarily agreeing to take part in the study. A special thank you to the student psychologist who conducted the assessments. Thank you Thandokuhle Nxiweni for assisting with the literature search.

**Conflicts of Interest:** The author declares no conflict of interest.

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
