# Peer review of "Obesity and Neurocognitive Performance of Memory, Attention, and Executive Function"

_neurosci, doi:10.3390/neurosci3030027_

Round 1
Reviewer 1 Report
The manuscript presented by Antonio G. Lentoor entitled "Obesity and Neurocognitive Performance of Memory, Attention and Executive Function" is interesting, original and well written. The results are clear and well described.
However I would suggest the author to insert a graphic summary of the procedure used to collect all data.
This will help the readers to understand this interesting study.
Author Response
Dear reviewer,
Thank you for taking the time to review this manuscript and providing valuable feedback to improve the quality of the paper.
Regards.

Reviewer 2 Report
The article is well written, the introduction very clear and the whole presentation is fine. I have few considerations:
1) how depression was investigated? There is a score in table 1, but it is not clear what it refers to.
2) authors may consider to study even the correlation between BMI and cognitive fields score using Pearson's correlation coefficient. It could be useful even to add a graph showing the distribution of BMI with the different scores. Data comparing even abdominal circumference and cognitive test would have been interesting too.
3) As a suggestion, it would have been useful to have data about sleep quality and possibile OSAS in obese patients.
4) there is no mention in the discussion about the possible influence of age and comorbidity differences between the two groups on the cognitive results.
5) Authors may consider to add the following references:
- Tang X, Zhao W, Lu M, Zhang X, Zhang P, Xin Z, Sun R, Tian W, Cardoso MA, Yang J, Simó R, Zhou JB, Stehouwer CDA. Relationship between Central Obesity and the incidence of Cognitive Impairment and Dementia from Cohort Studies Involving 5,060,687 Participants. Neurosci Biobehav Rev. 2021 Nov;130:301-313. doi: 10.1016/j.neubiorev.2021.08.028. Epub 2021 Aug 28. PMID: 34464646.
- Arnoldussen IA, Kiliaan AJ, Gustafson DR. Obesity and dementia: adipokines interact with the brain. Eur Neuropsychopharmacol. 2014;24(12):1982-1999. doi:10.1016/j.euroneuro.2014.03.002
Author Response
Dear reviewer,
Thank you for reviewing this manuscript an the valuable feedback that helped improve the quality of the paper.
Regards.

Round 2
Reviewer 1 Report
The author satisfied all my concerns. I propose the acceptance.